# Costs incurred by people with co-morbid tuberculosis and diabetes and their households in the Philippines

**Takuya Yamanaka**[1,2]*, **Mary Christine Castro**[3], **Julius Patrick Ferrer**[3], **Juan Antonio Solon**[3], **Sharon E. Cox**[2,4,5,6], **Yoko V. Laurence**[1,7], **Anna Vassall**[1]

**1** Department of Global Health and Development, London School of Hygiene and Tropical Medicine, London, United Kingdom, **2** School of Tropical Medicine and Global Health, Nagasaki University, Nagasaki, Japan, **3** Nutrition Center of the Philippines, Muntinlupa City, Philippines, **4** Faculty of Epidemiology and Population Health, London School of Hygiene and Tropical Medicine, London, United Kingdom, **5** Institute of Tropical Medicine, Nagasaki University (NEKKEN), Nagasaki, Japan, **6** UK Health Security Agency, London, United Kingdom, **7** Health Economics for Life Sciences and Medicine, Department of Population Health Sciences, King's College London, London, United Kingdom

* takuya.yamanaka@lshtm.ac.uk

**Data Availability Statement:** The survey dataset contains privacy-sensitive information including participant's individual and household income that formed a core part of the analysis. Even though we

## Abstract

### Objective

Diabetes is a risk factor for TB mortality and relapse. The Philippines has a high TB incidence with co-morbid diabetes. This study assessed the pre- and post-TB diagnosis costs incurred by people with TB and diabetes (TB-DM) and their households in the Philippines.

### Methods

Longitudinal data was collected for costs, income, and coping mechanisms of TB-affected households in Negros Occidental and Cebu, the Philippines. Data collection was conducted four times during TB treatment. The data collection tools were developed by adapting WHO's cross-sectional questionnaire in the *Tuberculosis Patient Cost Surveys*: *A Handbook* into a longitudinal study design. Demographic and clinical characteristics, self-reported household income, number of facility visits, patient costs, the proportion of TB-affected households facing catastrophic costs due to TB (>20% of annual household income before TB), coping mechanisms, and social support received were compared by diabetes status at the time of TB diagnosis.

### Results

530 people with TB were enrolled in this study, and 144 (27.2%) had TB-DM based on diabetes testing at the time of TB diagnosis. 75.4% of people with TB-DM were more than 45 years old compared to 50.3% of people with TB-only (p<0.001). People with TB-DM had more frequent visits for TB treatment (120 vs 87 visits, p = 0.054) as well as for total visits for TB-DM treatment (129 vs 88 visits, p = 0.010) compared to those with TB-only. There was no significant difference in the proportion of TB-affected households facing catastrophic costs between those with TB-DM (76.3%) and those with TB-only (68.7%, p = 0.691).

remove patient's identifiers such as patient number and name, there is still a possibility that those who are familiar with the project sites and beneficiaries may be able to identify participants and their households. The informed consent signed by all participants explicitly mentioned that only the research team have access to the dataset. Due to such ethical and confidentiality restrictions, the survey dataset will be made available only upon request. Please consider tmgh_jimu@ml.nagasaki-u.ac.jp which is the email address of administrative office of Nagasaki university - this is an institutional contact which was not involved in our study and can ensure a long-term data availability.

**Funding:** This study was funded by following three programme/grants: NU WISE Programme (Nagasaki University, "Doctoral Programme for World-leading Innovative and Smart Education" for Global Health, "Global Health Elite Programme for Building a Healthier World"); WHO's Joint WPR/TDR Small Grants Scheme for 2018-2019 on implementation research on infectious diseases of poverty; Foundation for Advanced Studies on International Development (FASID) Scholarship Program Assistance for Higher Education. The contents of this article are the responsibility of the authors and do not necessarily reflect the views of the sponsors.

**Competing interests:** The authors have declared that no competing interests exist.

## Conclusion

People with TB-DM in the Philippines face extensive health service use. However, this does not translate into substantial differences in the incidence of catastrophic cost. Further study is required to understand the incidence of catastrophic costs due to diabetes-only in the Philippines.

## Introduction

In 2015, the World Health Organization (WHO) set the End TB Strategy, aiming "to ensure that no family is burdened with catastrophic expenses due to TB by 2020" [1]. To capture the situation of TB-associated household costs and monitor the progress toward achieving this target, WHO supports countries to conduct baseline and periodic TB patient cost surveys [2]. Their guideline prescribes conducting a national TB patient cost survey using a feasible and affordable cross-sectional design to assess direct costs (for medications, consultations, hospitalization, transportation, accommodation, and supplements) and indirect costs (such as income loss) [5, 6].

Diabetes increases the risk of progressing to active TB disease and may increase the risk of poor TB treatment outcomes, thus contributing to ongoing transmission, particularly where diabetes is poorly managed [3–6]. As such, the risk of death and the risk of relapse are also higher among people with TB and diabetes (TB-DM) [4]. Therefore, enhancing diagnosis and management of diabetes may improve the rate of decline of TB incidence. The WHO published a separate guideline to develop and implement collaborative actions aimed at reducing the dual burden of TB-DM. The guideline included bi-directional screening of TB in people living with diabetes and of diabetes in people with TB; as well as monitoring and evaluation of collaborative TB-DM activities [7, 8].

Given this dual burden there is a concern that people with TB-DM may be more likely to suffer damaging levels of associated costs than those with TB alone, yet there is limited data available on how co-morbidity impacts household costs associated with TB. For HIV, another common comorbidity of TB, a study assessing patient costs for TB-only, HIV-only, and TB-HIV showed that people with TB-HIV visited health facilities more frequently (18.4 times per month) than those with TB-only (16.0 times) or HIV-only (2.2 times) due to fragmented services [9]. A similar situation is expected for the comorbidity of TB-DM, and therefore this study assesses the costs of diabetes diagnosis and treatment among people with TB-DM.

## Methods

### Study setting

The Philippines is classified by the WHO as one of the 30 high burden countries for both drug susceptible TB (DS-TB) and multidrug-resistant and rifampicin-resistant TB [10–12], with an estimated TB incidence of 650 per 100,000 in 2021 [11, 12]. In the Philippines, the National Tuberculosis Control Program (NTP) conducted a nationwide TB patient cost survey in 2015–2017 using the WHO recommended method [2, 13]. The results of the survey found 42.4% (95% confidence interval (95% CI) 40.2%-44.6%) of TB-affected households faced catastrophic costs [12, 14]. The Philippines has a high TB incidence with co-morbid diabetes (22,000 adult TB incident cases were attributable to diabetes in 2021) [12, 15].

In the Philippines, costs for diabetes diagnosis and management are not fully covered by national insurance, the NTP or the non-communicable disease control programme, but direct medical costs for TB treatment and diagnosis are covered [16, 17]. Social support is provided by the NTP for people with drug-resistant TB (DR-TB) with the purpose of improving treatment adherence and includes food packages and transportation fees for visiting health facilities [14, 18, 19]. Furthermore, The Department of Social Welfare and Development (DSWD) of the Philippines has a nationwide conditional cash transfer (CCT) programme for households living in poverty, and as of 2016, the CCT programme covered 4.4 million households, equivalent to 20% of the total population [20].

Patient costs for TB and diabetes were collected within an ongoing cohort study. The aim of the main cohort study was to measure the effects of malnutrition and diabetes on TB treatment outcomes in people with TB in Manila, Negros Occidental and Cebu, Philippines [21].

## Study design

We collected cost data from people with TB, enrolled in the cohort study between November 2018 and October 2020. The aim was to assess the changes in costs incurred, income earned and coping mechanisms before TB diagnosis until completion of TB treatment, and also assess the difference in costs incurred by people with TB-DM and TB-only. People with pulmonary TB including DR-TB who were 18 years and over were eligible to participate in this study following the eligibility criteria of the main study. Although people with TB-HIV were included in the main study, they were excluded in this costing study to exclude the financial impact from TB-HIV coinfection.

This costing study used a sub-sample of 11 health facilities located in two regions of the Philippines: Negros Occidental and Cebu (S1 Text). After excluding people with HIV (expected 3%), 620 people with TB from the main study were expected to be eligible for this study from November 2018 to February 2020. Assuming a 90% consent rate and 91% treatment completion rate, we expected to collect patient cost data from a total of 502 people with TB. Given 9–12% of the cohort were estimated to have diabetes (45–60 people) [22], we estimated that our sample size of 502 people with TB was sufficiently powered to detect a minimum 17% increase in total costs [23, 24].

Research nurses were based in each study site to recruit study participants from the main study into this study. The research nurses explained the purpose of this sub-study using a printed information sheet, in relevant local languages and English. People who agreed to participate in this research and signed the additional informed consent form were enrolled. Data collection for patient costs, household income and coping strategies was subsequently conducted by an in-person interview at each participant's home or by telephone during the period of COVID-19 lockdowns. To capture the total TB and diabetes-related costs incurred, patient interviews were conducted four times per patient; at the start of TB treatment for costs before TB diagnosis, at the end of the TB intensive phase for costs in TB intensive phase, and during the middle and end of the TB continuation phase for costs during the first half of the TB continuation phase and costs during the second half of the TB continuation phase. The enrolment of study participants was conducted from 11 November 2018 until 21 February 2020, and all the required follow-up data collection completed on 4 August 2020.

We used data collection tools derived from the national TB patient cost survey adapted for the Philippines [14] and used the same cost categories to estimate the costs incurred by those with TB-DM and with TB-only. These in turn are based on the WHO Tuberculosis Patient Cost Survey handbook [2, 25]. Costs consisted of direct medical costs (including consultation fees, drugs, screening and diagnostic tests, and hospitalization), direct non-medical costs

(including transportation, food and supplements, and accommodation), and indirect costs (income losses).

Costs per phase, which were collected at the start of TB treatment, at the end of the TB intensive phase, during the middle and at the end of the TB continuation phase, were interpolated using the data collected on costs at the last visit by visit types (i.e. hospitalization, directly observed therapy, medical follow-up and drug pick-up) and the frequency of each visit type during each phase. Then total costs were estimated by summing the costs per phase. Costs are considered as catastrophic when the total of direct medical, non-medical and indirect costs exceeded 20% of ability to pay (i.e. annual household income) following the WHO definition [2]. Reported annual household income before having TB was used as a primary indicator for ability to pay (output approach). The output approach is a measure of indirect cost and uses a difference in self-reported household income at each time point of data collection to estimate income changes during a TB episode. For TB-affected households reporting zero income before having TB, annual household income was imputed using a regression model based on household assets information (S1 Table).

Data cleaning and processing, statistical analyses, and data visualizations were performed using R4.2.0 (CRAN: Comprehensive R Archive Network). Mean with standard deviation (SD) and 95% CI, and median with interquartile range (IQR) were used for continuous data, and frequency with proportion (%) was presented for categorical data. All results were stratified by the diabetes status at the time of TB diagnosis. The diabetes screening (HbA1c and RPG: Random Plasma Glucose) and confirmatory (OGTT: Oral Glucose Tolerance Test) tests were provided in the main study. Those who had previously known diabetes, who had a blood sugar level >7.8 mmol/L by OGTT, and who had HbA1c >6.5% or RPG >200mg/dL at the time of TB diagnosis were categorized as people with TB and diabetes. Statistical differences between people with and without diabetes were tested using a chi-square test for categorical data and the t-test or Kruskal–Wallis test for continuous data. Statistical significance was defined as a p-value less than 0.05. Data on costs and incomes were collected in Philippine Pesos (Php) and later converted into United States dollars (USD) for analysis at the rate of Php 51.19 to USD 1 using the average UN Operational Rates of Exchange during the data collection period (November 2018-October 2020).

### Ethical considerations

A written consent form was obtained from each participant prior to enrolment, explicitly stating that only the principal investigator (PI) and co-PIs were able to access the study dataset. Prior to obtaining a written informed consent, our data collecotrs explained the purpose of this research with a written information sheet during patients' waiting time at each study site. Following ethics committees approved the consent procedure.

The St. Cabrini Medical Center-Asian Eye Institute Ethics Review Committee (SCMC-AEI ERC) reviewed and provided a Philippine national ethics approval for the main study, including approval for this sub-study (ERC #2018–008). Ethics approvals were also obtained from the Ethics Review Committee of the WHO Regional Office for the Western Pacific (Ref: 2019.18.PHL.4.STB) and Ethics Review Committees at the London School of Hygiene & Tropical Medicine and Nagasaki University.

## Results

### Study population

A total of 530 adults with TB were enrolled at TB diagnosis. Of these, 386 (72.8%) had TB only and 144 (27.2%) had TB-DM according to the status of the known diabetes, the OGTT and the

HbA1c or RPG tests at the time of TB diagnosis. Out of 144 participants who were categorized as TB-DM according to their status at the time of TB diagnosis, 48 (33.3%) knew their DM status, with 39 reported managing DM and 9 reported not managing DM. Most (79.4%) of the study participants completed TB treatment, while 15.6% had loss-to-follow-up, 1.2% had treatment failure, and 3.9% died during TB treatment. Of 530 participants, 445 completed every data collection point until the end of the continuation phase and were included in the analyses.

The majority (70.2%) of people with TB-DM were 45 years old or over, while only half of the people with TB-only were in that age group (p<0.001) (**Table 1**). The proportion with DR-TB was slightly higher amongst people with TB-DM (TB-DM: 15.7%, TB-only: 10.5%), but without a significant difference (p = 0.179). Higher BMI ($\geq$ 18.5) was observed amongst a greater proportion of people with TB-DM (TB-DM: 68.6%, TB-only: 52.6%, p = 0.004), and the proportion with a high blood glucose level (HbA1c $\geq$5.7) was also greater amongst people with TB-DM (TB-DM: 96.6%, TB-only: 35.0%, p<0.001).

## Health service utilisation

Mean total number of visits for TB-DM services amongst all the participants was 92.2 visits per person, and of these, 90.5 visits were for TB services and 1.8 for diabetes services (**Table 2**). People with TB-DM compared to people with TB-only, had more frequent visits for TB treatment (TB-DM: 120.0, TB-only: 86.9, p = 0.054) as well as more frequent aggregated visits for TB services and DM services (such as regular monitoring for blood sugar level and drug pickup) (TB-DM: 128.8, TB-only: 87.6, p = 0.010). There were no significant differences in the number of visits for TB services by treatment phase and by visit type between people with TB-DM and TB-only, except for medical follow-up (that is for physician's consultation and follow-up tests) between the middle and the end of the continuation phase (TB-DM: 1.1, TB-only: 0.7, p = 0.002).

## Costs incurred by TB-affected households

Overall, the mean total costs were estimated at USD 952 (**Table 3**). Of these, TB costs accounted for USD 932 (97.9%), and the TB costs were mainly driven by income loss (86.1%), followed by direct non-medical costs (10.5%) and direct medical costs (3.5%).

Although the total costs among people with TB-DM were USD 1,178, which was 28% higher than that incurred by people with TB only (USD 917), no significant difference was observed (p = 0.208). For TB services, while people with TB-DM incurred higher costs, there was no significant difference in TB costs between people with TB-DM and TB only (TB-DM: USD 1,053, TB only: USD 914, p = 0.464). For diabetes costs, people with TB-DM incurred significantly higher costs (TB-DM: USD 125) since people with TB-only incurred a minimal amount of costs (USD 4, p<0.001) for diabetes-related services, mainly for diabetes screening during TB treatment.

Among the three main cost categories for TB services (direct medical, direct non-medical, and income loss), a significant difference was shown only in direct non-medical costs between people with TB-DM and TB-only (TB-DM: USD 159, TB-only: USD 88, p<0.001), which was specifically for nutritional supplements and additional food (TB-DM: USD 87, TB-only: USD 40, p<0.001).

## Costs incurred by TB-DM affected households

People with TB and known diabetes that were already under management incurred much higher diabetes costs (TB-known and managed DM: USD 209, TB-unmanaged DM: USD 23,

**Table 1. Demographic and clinical characteristics of survey participants.**

| | TB patients without diabetes* | | TB patients with diabetes | | All TB patients | | P-value |
|---|---|---|---|---|---|---|---|
| | N | (%) | N | (%) | N | (%) | |
| **Total** | **324** | | **121** | | **445** | | |
| **Demographic characteristics** | | | | | | | |
| Sex | | | | | | | |
| Female | 92 | 28.4% | 37 | 30.6% | 129 | 29.0% | 0.738 |
| Male | 232 | 71.6% | 84 | 69.4% | 316 | 71.0% | |
| Age group | | | | | | | |
| 18–44 | 166 | 51.2% | 36 | 29.8% | 202 | 45.4% | <0.001 |
| ≥45 | 158 | 48.8% | 85 | 70.2% | 243 | 54.6% | |
| Education level | | | | | | | |
| No education/Primary | 114 | 35.2% | 34 | 28.1% | 148 | 33.3% | 0.164 |
| High school | 155 | 47.8% | 58 | 47.9% | 213 | 47.9% | |
| University or higher/Vocational | 55 | 17.0% | 29 | 24.0% | 84 | 18.9% | |
| Insurance status | | | | | | | |
| No insurance | 97 | 29.9% | 32 | 26.4% | 129 | 29.0% | 0.724 |
| PhilHealth | 138 | 42.6% | 56 | 46.3% | 194 | 43.6% | |
| GSIS/SSS | 89 | 27.5% | 33 | 27.3% | 122 | 27.4% | |
| Household size, median (interquartile range) | 5 (1–12) | | 4 (1–14) | | 5 (1–14) | | |
| Employment status before TB | | | | | | | |
| Employed (Formal) | 65 | 20.1% | 24 | 19.8% | 89 | 20.0% | 0.815 |
| Employed (Informal) | 127 | 39.2% | 51 | 42.1% | 178 | 40.0% | |
| Unemployed | 105 | 32.4% | 39 | 32.2% | 144 | 32.4% | |
| Student/Retired | 27 | 8.3% | 7 | 5.8% | 34 | 7.6% | |
| Primary income earner | | | | | | | |
| No | 170 | 52.5% | 57 | 47.1% | 227 | 51.0% | 0.368 |
| Yes | 154 | 47.5% | 64 | 52.9% | 218 | 49.0% | |
| **Clinical characteristics** | | | | | | | |
| Drug resistance status | | | | | | | |
| Drug susceptible-TB | 290 | 89.5% | 102 | 84.3% | 392 | 88.1% | 0.179 |
| Drug resistant-TB | 34 | 10.5% | 19 | 15.7% | 53 | 11.9% | |
| Treatment history | | | | | | | |
| New | 212 | 66.0% | 80 | 66.7% | 292 | 66.2% | 0.791 |
| Relapse | 98 | 30.5% | 36 | 30.0% | 134 | 30.4% | |
| Treatment after loss to follow up | 7 | 2.2% | 3 | 2.5% | 10 | 2.3% | |
| Treatment after failure | 3 | 0.9% | 0 | 0.0% | 3 | 0.7% | |
| Unknown | 1 | 0.3% | 1 | 0.8% | 2 | 0.5% | |
| Body mass index | | | | | | | |
| <18.5 (kg/m$^2$) | 153 | 47.4% | 38 | 31.4% | 191 | 43.0% | 0.004 |
| >= 18.5 (kg/m$^2$) | 170 | 52.6% | 83 | 68.6% | 253 | 57.0% | |
| Diagnostic delay (>4weeks) | 221 | 68.2% | 92 | 76.0% | 313 | 70.3% | - |
| Duration of TB episode (weeks) | | | | | | | |
| Care seeking: Mean (SD) | 11.3 | 17.5 | 11.2 | 13.4 | 11.3 | 16.5 | 0.942 |
| Intensive phase: Mean (SD) | 9.2 | 3.3 | 9.6 | 3.7 | 9.3 | 3.4 | 0.231 |
| Continuation phase: Mean (SD) | 16.8 | 2.4 | 17.0 | 2.6 | 16.9 | 2.4 | 0.526 |
| Hospitalized due to TB | 34 | 10.5% | 9 | 7.4% | 43 | 9.7% | 0.429 |
| Previously hospitalized in the current treatment phase | 255 | 78.7% | 90 | 74.4% | 345 | 77.5% | - |

*(Continued)*

**Table 1.** (Continued)

| | TB patients without diabetes* | | TB patients with diabetes | | All TB patients | | P-value |
|---|---|---|---|---|---|---|---|
| Treatment supports in intensive phase | | | | | | | |
| Self-administered | 241 | 74.4% | 83 | 68.6% | 324 | 72.8% | 0.271 |
| With treatment partner | 83 | 25.6% | 38 | 31.4% | 121 | 27.2% | |
| Treatment supports in middle of continuation phase | | | | | | | |
| Self-administered | 252 | 77.8% | 87 | 71.9% | 339 | 76.2% | 0.242 |
| With treatment partner | 72 | 22.2% | 34 | 28.1% | 106 | 23.8% | |
| Treatment supports in end of continuation phase | | | | | | | |
| Self-administered | 255 | 78.7% | 90 | 74.4% | 345 | 77.5% | 0.398 |
| With treatment partner | 69 | 21.3% | 31 | 25.6% | 100 | 22.5% | |
| HbA1c | | | | | | | |
| HbA1c:<5.7 | 207 | 65.9% | 4 | 3.3% | 211 | 48.6% | <0.001 |
| HbA1c:5.7–6.4 | 107 | 34.1% | 19 | 15.8% | 126 | 29.0% | |
| HbA1c:6.5+ or RPG:200+ (mg/dL) | 0 | 0.0% | 97 | 80.8% | 97 | 22.4% | |

**Table 2. Health service utilizations, mean per person.**

| | People with TB only | | People with TB and diabetes | | Overall | | p-value |
|---|---|---|---|---|---|---|---|
| | Mean | 95% CI | Mean | 95% CI | Mean | 95% CI | |
| **For TB services** | | | | | | | |
| **Before TB diagnosis** | **5.1** | **(4.9–5.3)** | **5.4** | **(4.8–5.9)** | **5.1** | **(4.9–5.3)** | **0.398** |
| **Intensive phase** | | | | | | | |
| Medical follow-up | 1.0 | (0.9–1.2) | 1.1 | (0.8–1.4) | 1.0 | (0.9–1.2) | 0.532 |
| Drug pickup | 16.8 | (13.5–20.2) | 23.0 | (12.4–33.7) | 17.5 | (14.3–20.7) | 0.156 |
| Directly observed therapy | 21.0 | (16.8–25.1) | 28.9 | (17.0–40.8) | 21.8 | (17.9–25.7) | 0.085 |
| **Middle of continuation phase** | | | | | | | |
| Medical follow-up | 0.6 | (0.5–0.6) | 0.7 | (0.4–0.9) | 0.6 | (0.5–0.6) | 0.289 |
| Drug pickup | 9.8 | (8.0–11.5) | 13.0 | (7.6–18.4) | 10.1 | (8.4–11.8) | 0.276 |
| Directly observed therapy | 12.7 | (10.1–15.4) | 18.6 | (11.1–26.0) | 13.4 | (10.9–15.9) | 0.114 |
| **End of continuation phase** | | | | | | | |
| Medical follow-up | 0.7 | (0.6–0.8) | 1.1 | (0.8–1.3) | 0.7 | (0.7–0.8) | 0.002 |
| Drug pickup | 8.0 | (6.6–9.4) | 11.7 | (6.5–16.9) | 8.4 | (7.0–9.8) | 0.178 |
| Directly observed therapy | 11.2 | (8.8–13.7) | 16.6 | (9.1–24.0) | 11.8 | (9.5–14.2) | 0.182 |
| **For diabetes services** | | | | | | | |
| **Intensive phase** | | | | | | | |
| Monitoring | 0.2 | (0.1–0.3) | 1.2 | (0.8–1.5) | 0.3 | (0.2–0.4) | <0.001 |
| Drug pickup | 0.2 | (0.06–0.3) | 2.5 | (1.6–3.3) | 0.5 | (0.3–0.6) | <0.001 |
| **Middle of continuation phase** | | | | | | | |
| Monitoring | 0.03 | (0.007–0.06) | 0.5 | (0.1–0.9) | 0.1 | (0.04–0.2) | 0.020 |
| Drug pickup | 0.2 | (0.08–0.2) | 2.4 | (1.4–3.3) | 0.5 | (0.3–0.6) | <0.001 |
| **End of continuation phase** | | | | | | | |
| Monitoring | 0.03 | (0.007–0.06) | 0.3 | (0.1–0.4) | 0.1 | (0.03–0.1) | 0.003 |
| Drug pickup | 0.1 | (0.05–0.2) | 2.0 | (1.4–2.7) | 0.4 | (0.2–0.5) | <0.001 |
| **Total** | | | | | | | |
| TB total | 86.9 | (73.7–100.1) | 120.0 | (76.9–163.2) | 90.4 | (77.7–103.2) | 0.054 |
| Diabetes total | 0.7 | (0.5–1.0) | 8.8 | (6.8–10.9) | 1.8 | (1.4–2.2) | 0.056 |
| Total TB and diabetes | 87.6 | (74.4–100.8) | 128.8 | (85.1–172.6) | 92.2 | (79.4–105.0) | 0.010 |

**Table 3. Detail of costs incurred per TB-affected households by TB treatment phase (mean, percentage, 95%CI), by diabetes status at the time of TB diagnosis.**

| TB-DM patient costs, US$ | | | People with TB only | | | People with TB and diabetes | | | Overall | | | p-value |
|---|---|---|---|---|---|---|---|---|---|---|---|---|
| | | | Mean | % | (95% CI) | Mean | % | (95% CI) | Mean | % | (95% CI) | |
| **Pre-TB diagnosis** | Direct medical costs | | 27.4 | 3.0% | (17.6–37.1) | 37.3 | 3.5% | (25.8–48.8) | 28.7 | 3.1% | (20.1–37.3) | 0.197 |
| | Direct non-medical costs | | 27.2 | 3.0% | (22.3–32.2) | 41.1 | 3.9% | (25.6–56.7) | 29.1 | 3.1% | (24.3–33.9) | 0.096 |
| | Income loss | | 205.7 | 22.5% | (148.1–263.4) | 308.1 | 29.3% | (86.2–530.1) | 219.3 | 23.5% | (161.2–277.4) | 0.382 |
| **Post-TB diagnosis** | Direct medical costs | Drug pickup | 0.06 | 0.0% | (0.0–0.1) | 0.0 | 0.0% | (0.0–0.0) | 0.05 | 0.0% | (0–0.1) | 0.101 |
| | | Directly observed therapy | 0.0 | 0.0% | (0.0–0.0) | 0.0 | 0.0% | (0.0–0.0) | 0.0 | 0.0% | (0.0–0.0) | N/A |
| | | Follow-up | 1.8 | 0.2% | (0.8–2.7) | 1.8 | 0.2% | (0.0–3.9) | 1.8 | 0.2% | (0.9–2.6) | 0.970 |
| | | Hospitalization | 1.9 | 0.2% | (0.0–4.8) | 0.8 | 0.1% | (0.0–2.4) | 1.8 | 0.2% | (0–4.3) | 0.527 |
| | | Total | 3.7 | 0.4% | (0.7–6.8) | 2.6 | 0.2% | (0.0–6.2) | 3.6 | 0.4% | (0.9–6.3) | 0.652 |
| | Direct non-medical costs | Accommodation | 0.05 | 0.0% | (0.0–0.1) | 0.1 | 0.0% | (0.0–0.3) | 0.06 | 0.0% | (0–0.1) | 0.632 |
| | | Food | 3.0 | 0.3% | (2.1–3.8) | 7.6 | 0.7% | (2.8–12.5) | 3.6 | 0.4% | (2.6–4.6) | 0.065 |
| | | Travel | 18.6 | 2.0% | (15.4–21.8) | 22.7 | 2.2% | (14.6–30.8) | 19.2 | 2.1% | (16.2–22.1) | 0.360 |
| | | Nutrition supplement | 39.5 | 4.3% | (34.0–45.0) | 87.4 | 8.3% | (65.1–109.7) | 45.8 | 4.9% | (40.0–51.7) | <0.001 |
| | | Total | 61.1 | 6.7% | (53.4–68.8) | 117.8 | 11.2% | (89.5–146.1) | 68.6 | 7.4% | (60.8–76.5) | <0.001 |
| | Income loss | | 588.3 | 64.4% | (489.7–687.0) | 545.9 | 51.8% | (350.2–741.6) | 582.7 | 62.5% | (493.3–672.1) | 0.704 |
| **Total direct medical costs** | | | 31.1 | 3.4% | (20.9–41.3) | 40.0 | 3.8% | (28.2–51.8) | 32.3 | 3.5% | (23.3–41.3) | 0.267 |
| **Total direct non-medical costs** | | | 88.4 | 9.7% | (78.4–98.3) | 158.9 | 15.1% | (123.1–194.7) | 97.7 | 10.5% | (87.6–107.8) | <0.001 |
| **Income loss** | | | 794.1 | 86.9% | (653.1–935.0) | 854.0 | 81.1% | (519.9–1 188.1) | 802.0 | 86.1% | (672.0–932.1) | 0.746 |
| **Total cost (TB)** | | | 913.5 | 100% | (768.8–1 058.3) | 1,052.9 | 100% | (709.3–1 396.4) | 932.0 | 100% | (798.4–1 065.7) | 0.464 |
| **For diabetes services** | | | | | | | | | | | | |
| **Direct medical costs** | | | 2.7 | 73.0% | (0.6–4.7) | 104.2 | 83.3% | (47.9–160.5) | 16.1 | 81.3% | (7.8–24.4) | <0.001 |
| **Direct non-medical costs** | | | 1.1 | 29.7% | (0.5–1.7) | 20.9 | 16.7% | (12.4–29.4) | 3.7 | 18.7% | (2.3–5.1) | <0.001 |
| **Total (Diabetes)** | | | 3.7 | 100% | (1.2–6.3) | 125.1 | 100% | (61.9–188.2) | 19.8 | 100% | (10.4–29.3) | <0.001 |
| **Total cost** | | | | | | | | | | | | |
| **Total cost (TB)** | | | 913.5 | 99.6% | (0.5–1.7) | 1,052.9 | 89.4% | (709.3–1 396.4) | 932.0 | 97.9% | (2.3–5.1) | 0.464 |
| **Total cost (Diabetes)** | | | 3.7 | 0.4% | (1.2–6.3) | 125.1 | 10.6% | (61.9–188.2) | 19.8 | 2.1% | (10.4–29.3) | <0.001 |
| **Total cost (TB-diabetes)** | | | 917.3 | 100% | (772.3–1 062.3) | 1,177.9 | 100% | (800.0–1 555.8) | 951.8 | 100% | (816.2–1 087.5) | 0.208 |

* Costs data was converted to United States Dollars (US$) from Philippines Peso (Php) using the average UN Operational Rates of Exchange during data collection period (Nov 2018-Oct 2020) of US$1 = Php 51.193375 (https://treasury.un.org/operationalrates/OperationalRates.ph

N/A: Not available

p<0.001), while there was no significant difference in the total TB-DM costs (TB-known and managed DM: USD 1363, TB-unmanaged DM: USD 841, p<0.078) (**Table 4**).

## Household income, catastrophic cost, and social support schemes

Overall, the mean reported monthly household income before having TB was USD 183 (95% CI: 155–210), with no significant differences between people with TB-DM (USD 189, 95%CI: 140–238) and TB-only (USD 182, 95%CI: 151–212), declining during TB diagnosis (USD 80, 95%CI: 68–92) and at the end of the intensive phase of TB treatment (USD 9, 95%CI: 6–11) (**Table 5**). It increased towards the middle of the continuation phase (USD 195, 95%CI: 164–

**Table 4. Detail of costs incurred per TB-DM affected households (mean, percentage, 95%CI), by diabetes management status at the time of TB diagnosis.**

| TB and diabetes patient costs, US$ | | | TB patients with known and managed diabetes* | | | TB patients with unmanaged diabetes* | | | p-value |
|---|---|---|---|---|---|---|---|---|---|
| | | | Mean | % | (95% CI) | Mean | % | (95% CI) | |
| **Pre-TB diagnosis** | Direct medical costs | | 34.7 | 3.0% | (19.9–49.4) | 27.6 | 3.4% | (19.4–35.8) | 0.412 |
| | Direct non-medical costs | | 44.6 | 3.9% | (21.7–67.5) | 26.0 | 3.2% | (17.6–34.3) | 0.138 |
| | Income loss | | 309.6 | 26.8% | (0.0–643.7) | 214.9 | 26.3% | (111.2–318.6) | 0.596 |
| **Post-TB diagnosis** | Direct medical costs | Drug pickup | 0.0 | 0.0% | (0.0–0.0) | 0.0 | 0.0% | (0.0–0.0) | N/A |
| | | Directly observed therapy | 0.0 | 0.0% | (0.0–0.0) | 0.0 | 0.0% | (0.0–0.0) | N/A |
| | | Follow-up | 2.8 | 0.2% | (0.0–6.4) | 1.0 | 0.1% | (0.2–1.8) | 0.355 |
| | | Hospitalization | 1.5 | 0.1% | (0.0–4.5) | 0.1 | 0.0% | (0.0–0.4) | 0.363 |
| | | Total | 4.3 | 0.4% | (0.0–10.8) | 1.2 | 0.1% | (0.3–2.0) | 0.350 |
| | Direct non-medical costs | Accommodation | 0.2 | 0.0% | (0.0–0.6) | 0.0 | 0.0% | (0.0–0.0) | 0.314 |
| | | Food | 9.8 | 0.8% | (1.2–18.4) | 3.2 | 0.4% | (1.7–4.6) | 0.137 |
| | | Travel | 28.2 | 2.4% | (14.6–41.8) | 19.4 | 2.4% | (14.2–24.6) | 0.240 |
| | | Nutrition supplement | 103.0 | 8.9% | (69.8–136.2) | 44.0 | 5.4% | (31.9–56.2) | 0.001 |
| | | Total | 141.2 | 12.2% | (98.0–184.5) | 66.6 | 8.1% | (51.9–81.4) | 0.002 |
| | Income loss | | 619.2 | 53.7% | (378.9–859.6) | 481.4 | 58.9% | (341.2–621.7) | 0.334 |
| **Total direct medical costs** | | | 39.0 | 3.4% | (23.7–54.3) | 28.7 | 3.5% | (20.2–37.3) | 0.254 |
| **Total direct non-medical costs** | | | 185.8 | 16.1% | (133.5–238.1) | 92.6 | 11.3% | (73.5–111.7) | 0.001 |
| **Income loss** | | | 928.8 | 80.5% | (492.3–1 365.4) | 696.3 | 85.2% | (484.8–907.9) | 0.349 |
| **Total cost (TB)** | | | 1,153.6 | 100% | (697.1–1 610.1) | 817.7 | 100% | (601.0–1 034.3) | 0.195 |
| **For diabetes services** | | | | | | | | | |
| **Direct medical costs** | | | 181.2 | 86.5% | (85.4–276.9) | 15.1 | 65.7% | (6.0–24.1) | 0.001 |
| **Direct non-medical costs** | | | 28.4 | 13.5% | (14.6–42.2) | 7.9 | 34.3% | (4.5–11.3) | 0.005 |
| **Total (diabetes)** | | | 209.6 | 100% | (102.0–317.2) | 23.0 | 100% | (11.4–34.5) | 0.001 |
| **Total cost** | | | | | | | | | |
| **Total cost (TB)** | | | 1,153.6 | 84.6% | (14.6–42.2) | 817.7 | 97.3% | (4.5–11.3) | 0.195 |
| **Total cost (diabetes)** | | | 209.6 | 15.4% | (102.0–317.2) | 23.0 | 2.7% | (11.4–34.5) | 0.001 |
| **Total cost (TB and diabetes)** | | | 1,363.2 | 100% | (832.2–1 894.2) | 840.7 | 100% | (619.1–1 062.2) | 0.078 |

*Study participants who reported known diabetes and also were already taking diabetes management at the study enrolment were categorized as TB patients with managed diabetes.

227) and was sustained to the end of that phase (USD 197, 95%CI: 165–228). No significant differences were observed for the mean reported monthly household income between people with TB-DM and TB-only except at the end of the intensive phase (TB-DM: USD 3 (95%CI: 1–5), TB-only: USD 10 (95%CI: 7–12), p<0.001).

In line with the changes in income, the proportion of households living below the international poverty line was greatest at the end of the intensive phase but with no statistically significant difference between people with TB-DM and TB-only.

The proportion of TB-affected households spending more than 20% of their annual household income on TB-related services was 69.0% (95%CI: 64.7–73.3%), and there was no statistically significant difference between people with TB-DM (68.7%, 95%CI: 64.0–73.3%) and TB-only (71.2%, 95%CI: 59.3–83.1%), with a p-value of 0.691 (**Fig 1**). Unsurprisingly, the proportion of households incurring costs greater than 20% of their annual household income for TB and DM-related services was higher for people with TB-DM (76.3%, 95%CI: 65.1–87.5%), while there was no significant difference compared to people with TB-only (p = 0.207).

**Table 5. Reported household income and social support received by TB-affected households.**

| | People with TB only | | People with TB and diabetes | | Overall | | p-value |
|---|---|---|---|---|---|---|---|
| | **Mean** | **95% CI** | **Mean** | **95% CI** | **Mean** | **95% CI** | |
| **Self-reported monthly household Income (in US$)** | | | | | | | |
| Before onset of TB symptoms | 181.8 | (151.3–212.3) | 188.9 | (140.3–237.5) | 182.7 | (155.4–210.0) | 0.810 |
| At the time of TB diagnosis | 79.1 | (66.6–91.6) | 88.2 | (51.5–124.8) | 80.3 | (68.4–92.1) | 0.647 |
| At the end of intensive phase | 9.5 | (6.7–12.2) | 2.9 | (0.6–5.3) | 8.6 | (6.2–11.0) | <0.001 |
| At the middle of continuation phase | 196.4 | (160.7–232.1) | 187.8 | (139.4–236.2) | 195.3 | (163.6–226.9) | 0.780 |
| At the end of continuation phase | 195.3 | (160.4–230.1) | 205.0 | (152.4–257.7) | 196.5 | (165.5–227.6) | 0.761 |
| | **%** | **95% CI** | **%** | **95% CI** | **%** | **95% CI** | |
| **Impoverishment: TB-affected households below international poverty line, percentage (95% CI)** | | | | | | | |
| Before onset of TB symptoms | 47.7 | (43.2–52.3) | 46.4 | (34.6–58.4) | 47.5 | (43.3–51.8) | 0.835 |
| At the time of TB diagnosis | 74.0 | (69.9–77.9) | 75.4 | (64.4–84.9) | 74.2 | (70.3–77.8) | 0.806 |
| At the end of intensive phase | 87.9 | (84.7–90.7) | 91.3 | (83.4–96.8) | 88.3 | (85.4–90.9) | 0.210 |
| At the middle of continuation phase | 35.4 | (31.0–39.8) | 33.3 | (22.6–45.0) | 35.1 | (31.1–39.2) | 0.678 |
| At the end of continuation phase | 33.8 | (29.6–38.2) | 30.4 | (20.0–42.0) | 33.4 | (29.4–37.5) | 0.482 |
| **Conditional cash transfer for poor** | | | | | | | |
| Before TB diagnosis | 16.8 | (13.4–20.9) | 10.2 | (4.6–21.2) | 16.0 | (12.8–19.7) | 0.194 |
| Intensive phase | 16.6 | (13.2–20.6) | 8.5 | (3.5–19.1) | 15.5 | (12.4–19.2) | 0.110 |
| Middle of continuation phase | 17.9 | (14.4–22.0) | 10.2 | (4.6–21.2) | 16.9 | (13.6–20.6) | 0.142 |
| End of continuation phase | 15.8 | (12.5–19.8) | 8.5 | (3.5–19.1) | 14.8 | (11.8–18.5) | 0.141 |
| **Social supports for TB people** | | | | | | | |
| Before TB diagnosis | 2.8 | (1.6–5.1) | 1.7 | (0.2–11.5) | 2.7 | (4.7–1.5) | 0.611 |
| Intensive phase | 12.7 | (9.7–16.4) | 16.9 | (9.2–29.0) | 13.3 | (16.8–10.4) | 0.370 |
| Middle of continuation phase | 15.3 | (12.0–19.2) | 16.9 | (9.2–29.0) | 15.5 | (19.2–12.4) | 0.743 |
| End of continuation phase | 14.0 | (10.9–17.8) | 18.6 | (10.5–30.9) | 14.6 | (18.2–11.6) | 0.347 |

Cash from the CCT programme was received by 16.0% (95%CI: 12.8–19.7%) of TB-affected households before TB diagnosis, and the proportion remained constant throughout TB treatment (**Table 5**). Similarly, the social support package was received by 13.3% (95%CI: 16.8–10.4%) of TB-affected households during the TB intensive phase and remained at the same level during the TB continuation phase. There was no significant difference in the proportion of households receiving the social support package between people with TB-DM and TB-only during TB treatment. Social consequences of TB were summarised in **S2 Table**.

## Discussion

We found high costs due to TB-DM, with an overall mean total cost of USD 952, and catastrophic costs in a high proportion of households (69%). We did not however find any significant difference in costs incurred or levels of catastrophic costs between those with TB-only and with TB-DM. Both groups were found to have similar levels of income before the start of the study and similar levels of income loss during TB treatment. While on average those with TB-DM incurred slightly more non-medical expenses and income loss, this was not substantially higher than those with TB-only. Those with TB-DM did face a substantially higher burden in terms of health care usage, but this did not translate into higher total costs given the limited overall cost of medical expenses and transport costs in our patient populations. Also, diabetes costs were much higher among those who were already receiving diabetes management at the time of TB diagnosis, while this also did not translate into higher total costs for TB-DM due to the limited number of samples in our study.

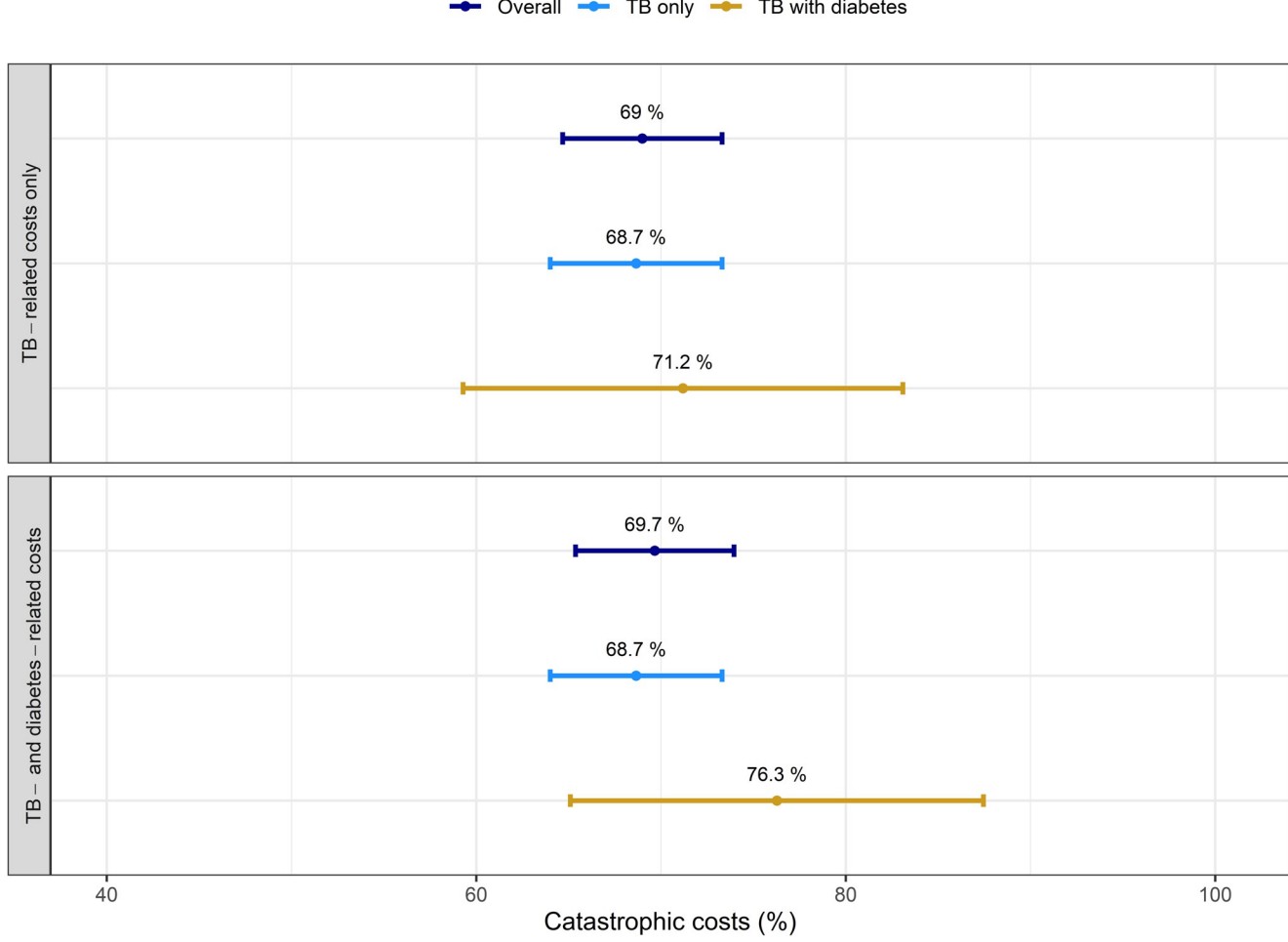

**Fig 1. Percentage of TB-affected households facing catastrophic costs > 20%.**

The unemployment rate of our study participants before having TB was already high at 34.5%, and approximately half (47%) of our participants were living under the international poverty line even before having TB. Those with TB-DM incurred a substantial amount of additional direct costs due to diabetes during the TB episode (USD 125). However, the total incurred costs for TB (USD 914 in TB-only, USD 1153 in TB-DM) had a far greater financial impact in affected households given their financial vulnerability due to high baseline unemployment and poverty rates. Therefore, in our study, the additional costs due to diabetes did not translate into a higher incidence of catastrophic costs and impoverishment during their TB episode.

This study was unable to capture costs for diabetes-related complications and hospitalizations in the sample of 144 people with TB-DM since the study was only assessing costs during a discrete period of a TB episode and not over the course of DM disease. Also, not all of them were taking DM management throughout the episode of TB. Therefore, our findings are not generalisable in describing the financial burden of diabetes. Lack of access to diabetes diagnosis and treatment usually result in the development of earlier, more frequent and severe complications such as blindness, kidney disease, coronary heart disease, cerebrovascular disease and stroke, and those complications lead to premature disability and death which incur a

higher financial burden in affected households [26, 27]. A previous study that assessed direct and indirect costs of diabetes in Kenya in 163 people showed that the total annual costs for diabetes services was USD 673, with 10% and 12% of study participants reporting costs for hospitalizations and irregular facility visits, respectively [28]. In that study, more than 50% of diabetes-affected households faced catastrophic costs (using a threshold of 20% of annual household income). Another study assessing 6,359 people in China showed that hospitalizations accounted for 73% of total diabetes costs, and the incidence of catastrophic costs was 24%, even with a higher threshold (40%) of annual household income [29]. Hence, another study with a larger sample size of people with diabetes is required to understand the entire picture of patient costs, incidence of catastrophic costs, and impoverishment due to diabetes in the Philippines.

In our study sites, integrated services for TB and diabetes were provided only in health facilities with programmatic management of drug resistant TB, and therefore most of the study participants with drug susceptible TB had to have separate facility visits for diabetes services. For example, the initial screening and regular monitoring for diabetes by point of care HbA1c or fasting blood glucose were not always provided in public health facilities, and therefore people living with diabetes had to visit private pharmacies and laboratories for these services. This study found that people with TB-DM had 40 extra visits to health facilities and/or treatment partners (e.g. facility and/or community DOT) compared with those with TB only. Therefore, the reduction in visits to healthcare providers and related costs (e.g. travel and food costs in direct non-medical costs) might be achieved by integrated care for TB and diabetes. However, given the high incidence of catastrophic costs regardless of diabetes status in this study, it is unlikely that catastrophic costs can be mitigated only by ensuring the health service integration.

TB-affected households in this study lost almost 95% of their monthly household income at the end of the intensive phase, and this highlighted that TB-affected households may become financially vulnerable and require social and/or financial support during the TB intensive phase. However, only around 15% of the participants received financial support from the nationwide CCT programme for households living under poverty. It did not increase throughout TB treatment, even though their household income was considerably reduced at the time of TB diagnosis and at the end of the TB intensive phase. A similar situation was observed in the national TB patient cost survey in the Philippines, however, an even lower proportion of survey participants (1.3%) were receiving the nationwide CCT programme provided by the DSWD of the Philippines [12, 14]. The national survey recommended that enhanced cooperation between NTP and DSWD is necessary for TB-affected households to benefit from financial support from the CCT programme. Our findings support the findings and recommendation from the national survey, and timely social protection and support are indispensable to avert catastrophic costs among TB-affected households in the Philippines.

This study had several limitations. First, it was conducted at 11 health facilities located in urban (Cebu) and rural (Negros) settings in the Philippines, and therefore, the results and findings cannot be generalized. Second, this study was able to enrol only a small sample of people with TB-DM (N = 144). Thus, further studies with a larger sample that assesses the financial impact of TB-DM is necessary. Third, although the longitudinal study design allowed multiple interviews during a TB episode with less recall bias compared to a cross sectional study, this study assessed costs from the onset of TB symptoms until the completion of TB treatment. Therefore, costs due to TB-related sequelae and/or prolonged social consequences after TB treatment were not investigated in this study. Fourth, approximately 15% of the enrolled participants were not able to complete all the data collection points due to dropout either from our study or TB from treatment. Therefore, results of catastrophic cost estimates

might be affected by attrition bias. Fifth, the costs were estimated from participants who completed interviews for four times and the sample size was N = 445, which did not reach the intended sample size of N = 502 due to unexpectedly high proportion of loss-to-follow-up (15.6%). Therefore, our sample size might be not powered enough to detect cost differences between TB-DM and TB-only.

## Conclusion

People with TB-DM in the Philippines face extensive health service use and incur higher costs to receive diabetes related health services. However, this does not translate into substantial differences in the incidence of catastrophic cost due to the baseline poverty in TB-affected households. Further study is required to understand the incidence of catastrophic costs due to diabetes-only in the Philippines.

## Supporting information

**S1 Text. List of study sites.**
(DOCX)

**S1 Table. Regression analysis for asset-based imputed household income.**
(DOCX)

**S2 Table. Social consequences of TB.**
(DOCX)

## Acknowledgments

We first would like to thank the people with TB who consented to participate in this study in the Philippines. Also, we acknowledge the contribution of the research nurses, Ms Bliss Craig and Ms Michelley Caballero, in Nutrition Center of the Philippines.

## Author Contributions

**Conceptualization:** Takuya Yamanaka, Sharon E. Cox, Yoko V. Laurence, Anna Vassall.

**Data curation:** Takuya Yamanaka.

**Formal analysis:** Takuya Yamanaka.

**Funding acquisition:** Takuya Yamanaka, Sharon E. Cox.

**Investigation:** Takuya Yamanaka, Mary Christine Castro, Julius Patrick Ferrer.

**Methodology:** Takuya Yamanaka, Yoko V. Laurence, Anna Vassall.

**Project administration:** Mary Christine Castro, Julius Patrick Ferrer.

**Resources:** Takuya Yamanaka.

**Software:** Takuya Yamanaka.

**Supervision:** Yoko V. Laurence, Anna Vassall.

**Validation:** Takuya Yamanaka.

**Visualization:** Takuya Yamanaka.

**Writing – original draft:** Takuya Yamanaka.

**Writing – review & editing:** Takuya Yamanaka, Mary Christine Castro, Julius Patrick Ferrer, Juan Antonio Solon, Sharon E. Cox, Yoko V. Laurence, Anna Vassall.

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
