## [Decision Letter · Decision Letter 0]

4 Dec 2023

PONE-D-23-20372Costs incurred by people with co-morbid tuberculosis and diabetes and their households in the PhilippinesPLOS ONE

Dear Dr. YAMANAKA,

Thank you for submitting your manuscript to PLOS ONE. After careful consideration, we feel that it has merit but does not fully meet PLOS ONE’s publication criteria as it currently stands. Therefore, we invite you to submit a revised version of the manuscript that addresses the points raised during the review process.

We look forward to receiving your revised manuscript.

Kind regards,

Lisa Kawatsu, PhD

Academic Editor

PLOS ONE

Journal Requirements:

3. Please include a complete copy of PLOS’ questionnaire on inclusivity in global research in your revised manuscript. Our policy for research in this area aims to improve transparency in the reporting of research performed outside of researchers’ own country or community. The policy applies to researchers who have travelled to a different country to conduct research, research with Indigenous populations or their lands, and research on cultural artefacts. The questionnaire can also be requested at the journal’s discretion for any other submissions, even if these conditions are not met.  Please find more information on the policy and a link to download a blank copy of the questionnaire here: https://journals.plos.org/plosone/s/best-practices-in-research-reporting. Please upload a completed version of your questionnaire as Supporting Information when you resubmit your manuscript.

Reviewers' comments:

Reviewer's Responses to Questions

**Comments to the Author**

1. Is the manuscript technically sound, and do the data support the conclusions?

Reviewer #1: Yes

Reviewer #2: Yes

2. Has the statistical analysis been performed appropriately and rigorously? 

Reviewer #1: Yes

Reviewer #2: Yes

3. Have the authors made all data underlying the findings in their manuscript fully available?

Reviewer #1: Yes

Reviewer #2: No

4. Is the manuscript presented in an intelligible fashion and written in standard English?

Reviewer #1: Yes

Reviewer #2: Yes

5. Review Comments to the Author

Reviewer #1: It is well documented that people with TB face financial hardship. This study is another important piece of research in documenting that – and goes even further by investigating costs for people with TB and diabetes co-morbidity.

The manuscript is well written, clear and concise. I have only the following minor points for clarification:

1. The time of recruitment of participants is not clear in the methods. In line 102-3 you say that interviews were conducted four times, with the first being at the start of TB treatment but given that you have pre-diagnosis utilization (Table 2) and costs (table , it would be good to know how long before the first interview had people been recruited and/or had TB symptoms and is there a possibility of recall bias?

2. It would be helpful to have a breakdown of the diabetes patients ie how many had previously known DM and how many were newly diagnosed at time of TB diagnosis and then some discussion as to whether this would have any cost implications? It also appears that many of the TB-DM patients had uncontrolled DM (Table 1 HbA1c result – which by the way is not clear when this result was?? Was it at the time of TB diagnosis? Please clarify)

3. In your sample size calculation you expected only 9-12% with DM (line 93) but in results (line 154) – this was 27.2%. Please explain.

4. Table 3: It is non-sensical to have a p-value when results are 0 as in Directly observed therapy

5. Table 1: Treatment history – was this before recruitment into the study? There is a high proportion of treatment relapse. Also – there was a higher % of TB-DM who were drug-resistant. Can you discuss what implications if any on costs?

6. Could the reason for no difference in cost between TB-only and TB-DM be because the DM patients were actually not getting the DM treatment they needed?? Ie only mean of 8.8 health service utilisations for DM (Table 2)

Reviewer #2: The manuscript reports the findings of assessment of the costs associated with TB and diabetes (TB-DM) in the Philippines.

The Philippines has a high incidence of TB with co-morbid diabetes. The data on the impact of the diabetes co-morbidity on the cost associated with TB is limited. The dual burden of TB and diabetes could significantly increase this cost in the countries where costs for diabetes diagnosis and management are not fully covered by national insurance.

The authors assessed the pre- and post-TB diagnosis costs incurred by the households of persons with TB-DM and TB only from 11 facilities of 2 regions of the Philippines. To estimate these costs, the authors used the methodology proposed by the WHO and available in the Tuberculosis Patient Cost Survey handbook, as well as the WHO definition of catastrophic cost due to TB (20% of ability to pay).

The data was collected withing ongoing parent cohort study aimed to assess the impact of malnutrition and diabetes on treatment outcomes in people with TB in three regions of the Philippines.

The authors used means (with SD and 95%CI) and medians (with IQR) to describe continuous data and frequencies (with %) for categorical data. Statistical differences were tested using the t-test or Kruskal–Wallis test for continuous data and the chi-square test for categorical data.

The authors found that people with TB and diabetes incur higher costs compared to those with TB-only. Although authors did not find statistically significant difference between the proportion of the households experienced catastrophic costs due to TB between those with TB-DM and TB-only, the study itself is very important as contributes to limited data available on the cost associated with TB-DM in the Philippines.

The manuscript is well written, the goal is well justified, public health message is clear, strong, and coherent with the data presented. The study dataset contains privacy-sensitive information including participant’s individual and household income, so it can be available only upon request.

There are a few issues to be addressed prior to publishing the manuscript. Those issues with the recommendations for improvement are listed below.

Comments

1. The authors enrolled 530 participants, but only 445 were included in the analysis; 85 (16%) were excluded (line 163). Socio-demographic characteristics of excluded 85 persons are not provided, as well as proportion of those with TB-DM among them.

-Table 1 includes 530 participants, but it seems that analysis results in all other tables (2-4) are provided for 445 participants. This is not clear from those tables. Please add size of analytic dataset to the titles of all tables in the manuscript. Also, please add the number of participants with TB-DM and TB-only in the titles of correspondent columns.

-In Table 1 please provide characteristics of 445 participants included in analysis or provide evidence of no difference between those 85 excluded and 445 included participants.

-As the analytic dataset included 445 participants (not 530), the sample size (502) was not reached, and power of the study was affected. The correspondent paragraph of the Methods section (lines 89-95) should be revised. This limitation should also be included in the Discussion section.

2. It is not clear how participants in 11 facilities were selected from the parent study to this study. Was it random, or all participants of the parent study were enrolled (according to the enrollment criteria) during indicated period (November 2018 to February 2020)?

3. The authors emphasized several times that the data was collected for persons with TB-DM, ignoring those with TB-only, however majority of enrolled persons had TB-only (386/530, 72.8%) (line 154). Please revise statements in lines 7, 75 and 81 to include data collection for participants with TB-only.

6. PLOS authors have the option to publish the peer review history of their article (what does this mean?). If published, this will include your full peer review and any attached files.

Reviewer #1: No

Reviewer #2: No

---

## [Author Response · Author response to Decision Letter 0]

12 Dec 2023

Thank you for your valuable comments and suggestions. We added our responses to each comment with red text and revised the manuscript accordingly.

Comments to the Author

Reviewer #1: 

It is well documented that people with TB face financial hardship. This study is another important piece of research in documenting that – and goes even further by investigating costs for people with TB and diabetes co-morbidity.

The manuscript is well written, clear and concise. I have only the following minor points for clarification:

1. The time of recruitment of participants is not clear in the methods. In line 102-3 you say that interviews were conducted four times, with the first being at the start of TB treatment but given that you have pre-diagnosis utilization (Table 2) and costs (table , it would be good to know how long before the first interview had people been recruited and/or had TB symptoms and is there a possibility of recall bias?

The timing of the recruitment was at the time of TB diagnosis. In the first data collection at the time of TB diagnosis (or within 7 days after the diagnosis), we collected the data for the duration from the time when participants started having TB-related symptoms (e.g. cough, weight loss etc) until the TB diagnosis, and the mean duration was 11 weeks with no significant difference between with and without diabetes (p=0.833). Considering this comment, we added this into table 1 – please see “Duration of TB episode: Care seeing” in updated table 1.

Regarding recall bias, yes it might exist, but our study design minimized the effect with the longitudinal study design, compared to the WHO recommended cross-sectional design which enrol those who are already on TB treatment – either in the TB intensive or continuation phase. 

Also, it is inherent unless a study is designed to enrol general population and follow up to see incidence of TB (such as TB prevalence survey) which will require massive amount of research budget. 

2. It would be helpful to have a breakdown of the diabetes patients ie how many had previously known DM and how many were newly diagnosed at time of TB diagnosis and then some discussion as to whether this would have any cost implications? It also appears that many of the TB-DM patients had uncontrolled DM (Table 1 HbA1c result – which by the way is not clear when this result was?? Was it at the time of TB diagnosis? Please clarify)

First, HbA1c test was provided for all participants at the time of TB diagnosis. 

Second, out of 144 participants who were categorized as TB-DM according to their status at the time of TB diagnosis, 48 (33.3%) knew their DM status – 39 reported managing DM and 9 reported not managing DM. We added this text in result section (line 156-158). As a total of 9 reported not managing DM and 96 people did not know their DM status, 105 were uncontrolled DM.

Cost differences by controlled vs uncontrolled DM are summarized and added in Result section as table 4, since those with controlled DM incurred much higher DM costs (TB-known and managed DM: USD 209, TB-unmanaged DM: USD 23, p<0.001). However, there was there was no significant difference in the total TB-DM costs (TB-known and managed DM: USD 1363, TB-unmanaged DM: USD 841, p<0.078) due to the limited number of samples in our study.

3. In your sample size calculation you expected only 9-12% with DM (line 93) but in results (line 154) – this was 27.2%. Please explain.

From a previous but recent study that assessed diabetes status in TB patients in the Philippines, we expected 9-12% in our sample size calculation. The difference compared to the result (27%) could be a because of different study population. The previous study was conducted in Metro Manila and Negros Occidental, while this study was in Cebu and Negros Occidental.

4. Table 3: It is non-sensical to have a p-value when results are 0 as in Directly observed therapy

Thank you for spotting this error. We updated the table replacing the p-value with “N/A”.

5. Table 1: Treatment history – was this before recruitment into the study? There is a high proportion of treatment relapse. Also – there was a higher % of TB-DM who were drug-resistant. Can you discuss what implications if any on costs?

Yes, the treatment history is for the previous TB histories before the recruitment into the study.

We found that DR-TB patients incurred higher TB-related costs (DS-TB: USD 861 vs USD 1456). However, since the comparison by drug-resistance status is not the focus of this manuscript, we did not include it. Also, the % of DR-TB did not translate into the higher costs when it’s compared by TB-DM vs TB only (p=0.464). Also, after revising table 1 with N=445 only, considering a comment from reviewer 2, there is no statistical significance in the drug-resistance status (p=0.179) among those who completed 4 times interviews.

6. Could the reason for no difference in cost between TB-only and TB-DM be because the DM patients were actually not getting the DM treatment they needed?? Ie only mean of 8.8 health service utilisations for DM (Table 2)

We interviewed (not as data collection) several people with TB-DM, and the required frequency for DM management was 1-2 times per month, and therefore 8.8 times of facility visits during TB treatment can be reasonable since the standard duration of DS-TB treatment is 6 months.

The reason for no difference in costs between TB-DM and TB-only is rather because in our study, no participants with TB-DM had DM-related complications or hospitalizations. As we stated in discussion section, in studies in Kenya and China that assessed DM patient costs, people with DM incurred much higher costs for DM due to DM-related complications and hospitalizations. Therefore, further studies will be required assessing costs for TB and DM-related complications. 

 

Reviewer #2: 

The manuscript reports the findings of assessment of the costs associated with TB and diabetes (TB-DM) in the Philippines.

The Philippines has a high incidence of TB with co-morbid diabetes. The data on the impact of the diabetes co-morbidity on the cost associated with TB is limited. The dual burden of TB and diabetes could significantly increase this cost in the countries where costs for diabetes diagnosis and management are not fully covered by national insurance.

The authors assessed the pre- and post-TB diagnosis costs incurred by the households of persons with TB-DM and TB only from 11 facilities of 2 regions of the Philippines. To estimate these costs, the authors used the methodology proposed by the WHO and available in the Tuberculosis Patient Cost Survey handbook, as well as the WHO definition of catastrophic cost due to TB (20% of ability to pay).

The data was collected withing ongoing parent cohort study aimed to assess the impact of malnutrition and diabetes on treatment outcomes in people with TB in three regions of the Philippines.

The authors used means (with SD and 95%CI) and medians (with IQR) to describe continuous data and frequencies (with %) for categorical data. Statistical differences were tested using the t-test or Kruskal–Wallis test for continuous data and the chi-square test for categorical data.

The authors found that people with TB and diabetes incur higher costs compared to those with TB-only. Although authors did not find statistically significant difference between the proportion of the households experienced catastrophic costs due to TB between those with TB-DM and TB-only, the study itself is very important as contributes to limited data available on the cost associated with TB-DM in the Philippines.

The manuscript is well written, the goal is well justified, public health message is clear, strong, and coherent with the data presented. The study dataset contains privacy-sensitive information including participant’s individual and household income, so it can be available only upon request.

There are a few issues to be addressed prior to publishing the manuscript. Those issues with the recommendations for improvement are listed below.

Comments

1. The authors enrolled 530 participants, but only 445 were included in the analysis; 85 (16%) were excluded (line 163). Socio-demographic characteristics of excluded 85 persons are not provided, as well as proportion of those with TB-DM among them.

-Table 1 includes 530 participants, but it seems that analysis results in all other tables (2-4) are provided for 445 participants. This is not clear from those tables. Please add size of analytic dataset to the titles of all tables in the manuscript. Also, please add the number of participants with TB-DM and TB-only in the titles of correspondent columns.

-In Table 1 please provide characteristics of 445 participants included in analysis or provide evidence of no difference between those 85 excluded and 445 included participants.

-As the analytic dataset included 445 participants (not 530), the sample size (502) was not reached, and power of the study was affected. The correspondent paragraph of the Methods section (lines 89-95) should be revised. This limitation should also be included in the Discussion section.

Thank you for this suggestion. We revised the first paragraph of Result section with brief description/characteristics (such as DM status and treatment outcomes) for all enrolled TB patients (N=530), and then described further details of characteristics of participants who completed 4-times interviews (N=445). Table 1 has been updated with N=445 accordingly.

2. It is not clear how participants in 11 facilities were selected from the parent study to this study. Was it random, or all participants of the parent study were enrolled (according to the enrollment criteria) during indicated period (November 2018 to February 2020)?

We reached all eligible participants of the parent study who were enrolled during the indicated period, and those who provided consent to participate in this study were enrolled into this study – that were N=530.

3. The authors emphasized several times that the data was collected for persons with TB-DM, ignoring those with TB-only, however majority of enrolled persons had TB-only (386/530, 72.8%) (line 154). Please revise statements in lines 7, 75 and 81 to include data collection for participants with TB-only.

Thank you for this suggestion. We revised the texts accordingly.

---

## [Editor Report · Decision Letter 1]

4 Jan 2024

Costs incurred by people with co-morbid tuberculosis and diabetes and their households in the Philippines

PONE-D-23-20372R1

Dear Dr. YAMANAKA,

We’re pleased to inform you that your manuscript has been judged scientifically suitable for publication and will be formally accepted for publication once it meets all outstanding technical requirements.

Kind regards,

Lisa Kawatsu, PhD

Academic Editor

PLOS ONE

---

## [Editor Report · Acceptance letter]

14 Jan 2024

PONE-D-23-20372R1 

PLOS ONE

Dear Dr. Yamanaka, 

I'm pleased to inform you that your manuscript has been deemed suitable for publication in PLOS ONE. Congratulations! Your manuscript is now being handed over to our production team.

Kind regards, 

on behalf of

Dr. Lisa Kawatsu 

Academic Editor

PLOS ONE